# Multiple Chromatographic Analysis of Urine in the Detection of Bladder Cancer

**DOI:** 10.3390/diagnostics11101793

**Published:** 2021-09-28

**Authors:** Daniela Džubinská, Milan Zvarík, Boris Kollárik, Libuša Šikurová

**Affiliations:** 1Department of Nuclear Physics and Biophysics, Faculty of Mathematics, Physics and Informatics, Comenius University in Bratislava, Mlynská Dolina, 842 48 Bratislava, Slovakia; daniela.dzubinska@fmph.uniba.sk (D.D.); libusa.sikurova@fmph.uniba.sk (L.Š.); 2Department of Urology, University Hospital of Bratislava, Antolská 11, 851 07 Bratislava, Slovakia; bkollarik@gmail.com

**Keywords:** bladder cancer, biomarker, non-malignant hematuria, 3D HPLC

## Abstract

Bladder cancer (BC) is the most common type of carcinoma of the urological system. Recently, there has been an increasing interest in non-invasive diagnostic tumor markers due to the invasive attribute of cystoscopy, which is still considered the gold standard diagnostic method. However, markers published in the literature so far do not meet expectations for replacing cystoscopy due to their low specificity and excessively high false-positive results, which can be mainly caused by frequently occurring hematuria also in benign cases. No reliable non-invasive method has yet been identified that can distinguish patients with bladder cancer and non-malignant hematuria patients. Our work examined the possibilities of non-targeted biomarkers of urine to distinguish patients with malignant and non-malignant diseases of the bladder using 3D HPLC in combination with computer processing of multiple datasets. Urine samples from 47 patients, 23 patients with bladder cancer (BC) and 24 patients with non-malignant hematuria (NMHU), were enrolled in clinical trials. For the separation and subsequent analysis of a large number of urine components, 3D HPLC (high-performance liquid chromatography) with an absorption and fluorescence detector was used. The obtained dataset was further subjected to various uni- and multi-dimensional statistical analyses and mathematical modeling. We found 334 chromatographic peaks, of which 18 peaks were identified as significantly different for BC and NMHU patients. Using receiver operating characteristic (ROC) analysis, we assessed the informative ability of significant chromatographic peaks (90% sensitivity and 74% specificity). By logistic regression, we identified the optimal and simplified set of seven chromatographic peaks (5 absorptions plus 2 fluorescence) with strong classification power (100% sensitivity and 100% specificity) for distinguishing patients with bladder cancer and those with non-malignant hematuria. Partial least square discriminant analysis (PLS-DA) model and orthogonal projection to latent structure discriminant analysis (OPLS-DA) with 100% sensitivity and 96% specificity were used to distinguish BC and NMHU patients. Multivariate statistical analysis of urinary metabolomic profiles of patients revealed that BC patients can be discriminated from NMHU patients and the results can likely contribute to an early and non-invasive diagnosis of BC.

## 1. Introduction

Bladder cancer (BC) is one of the most common genitourinary malignancies. Additionally, it is the sixth most common cancer in men [1]. In most cases, BC is detected at a late stage, which represents an unfavorable prognosis for the patient. One reason for delayed diagnosis is the non-specificity of symptoms, such as difficulty with micturition, pain in urination, and blood in urine, which may be accompanied by different diseases unrelated to malignant tumors. Currently, cystoscopy and cytology are the gold standard methods for BC detection. Cystoscopy is considered an invasive and painful examination of BC [2]. This examination method often represents a physical burden on the patient [3]. Although cytology is a non-invasive method, its sensitivity is not sufficient (less than 40%), particularly for low-grade tumors [4]. Recently, research interest has shifted to non-invasive methods of identifying the early stages of BC [5].

Biomarkers have the potential to aid in diagnosis, surveillance, staging, prognosis, and possibly therapeutic guidance. A large number of potential biomarkers for the detection of genomic, transcriptomic, epigenetic, or protein changes in serum or urine samples have been described in the literature, but only some of them are approved by the Food and Drug Administration [2,5,6]. The most common are NMP22^®^ BladderChek^®^ Test, TRAK/BTA stat^®^ Test, UroVysion^®^ FISH, ImmunoCyt™/uCyt+™, and CxBladder, which often provide relatively low sensitivity [2]. For the time being, there is no single urinary biomarker achieved from non-invasive BC surveillance tests to replace cystoscopy; therefore, obtaining a diagnostic model that can distinguish BC patients from the others would be more beneficial [7].

The metabolome reflects the status of the biological system. Urine is in direct contact with bladder epithelial cells and metabolites released into the urine provide information on bladder disorders, suggesting that analysis of urinary metabolomic profile is also a promising approach for discrimination between malignant and benign diseases [8,9]. However, there has been limited metabolomic research in the detection of biomarkers specific for BC [10]. Hematuria is the most common presenting symptom in bladder cancer (present in 85% of cases) [11]. Nevertheless, only a few studies have targeted false-positive values caused by bleeding with a benign origin [9]. No reliable method has yet been identified that can distinguish patients with bladder cancer (BC) and non-malignant hematuria patients (NMHU).

In the present study, an untargeted metabolomics approach using 3D reverse-phase high-performance liquid chromatography (RP-HPLC) with absorption and fluorescence detection was carried out as a promising alternative to omics methods for searching for biomarkers in patient’s urine. By subsequent computer processing of acquired extensive data (logistic regression, receiver operating characteristic (roc) analysis, partial least square discriminant analysis, orthogonal projection to latent structure discriminant analysis), we created classification models for the differentiation of patients with BC and NMHU.

## 2. Materials and Methods

### 2.1. Patients and Sample Collection

Urine samples were collected from 23 male patients with bladder cancer (BC) (median age of 65) and 24 male patients with non-malignant hematuria (NMHU) (median age of 65) (Table 1). Urine was collected by spontaneous miction or by catheterization. Collected samples were kept frozen at −28 °C. Each urine sample was thawed at room temperatures and centrifuged at 3000× *g* for 5 min, followed by filtration through a 0.22 µm nylon membrane filter-LLG Syringe Filter PTFE (AZ chrome s.r.o., Bratislava, Slovak Republic). Urine samples were provided by the Urological department UNB Saints Cyril and Methodius Hospital in Bratislava. Informed consent was signed by each patient and the project was approved by the ethics commission of UNB Saints Cyril and Methodius Hospital in Bratislava. Each patient underwent a biopsy, consequently, histological examination confirmed or excluded a malignant tumor of the bladder (diagnosis C67 according to the International classification of diseases MKCH 10). Based on that, the patient was classed in the appropriate group (BC or NMHU patients). Table 1 shows the basic characteristics of samples with the TNM (tumor–node–metastasis) staging system, describing the stage and grade of BC.

### 2.2. Principle of HPLC

High-performance liquid chromatography (HPLC) is an analytical method used to separate and quantify components from the sample due to differences in their structure. The apparatus consists of multiple modules. Mobile phase is pressed under high pressure into the column with stationary phase. A liquid sample is injected to the stream of mobile phase flowing through a column, whose function is to separate the compounds. Based on the affinity to the stationary phase, the components are eluted at different retention times. Retention time is specific for every compound. Components that display a stronger interaction with the stationary phase will be eluted later from the column, while components with weaker interaction will elute sooner. Subsequently, these eluates consecutively flow through one or more detectors, which generate chromatograms—intensity (voltage response) as function of time. The single chromatographic peaks represent a compound or group of compounds with similar characteristics. The area under the peaks represents the quantity of the compound [12]. 

### 2.3. Chemicals

Potassium dihydrogen phosphate (KH2PO4), water, acetonitrile (ACN), and methanol for HPLC were acquired from Honeywell, Regen, Germany. Sodium hydroxide (NaOH) was obtained from Slavus s.r.o., Bratislava, Slovak republic. HPLC creatinine standard was acquired from Sigma-Aldrich, Steinheim, Germany.

### 2.4. Equipment and Conditions

The urine samples were analyzed using reverse-phase high-performance liquid chromatography (RP-HPLC) system Prominence 20 A (Shimadzu Co., Kyoto, Japan) with absorption and fluorescence detection. The system temperature was adjusted to 25 °C. Nucleosil 100-5 C18 (150 × 4.6 mm; 5 µm particle size) (Macherey-Nagel, Düren, Germany) as the stationary phase and 10 mmol/L KH_2_PO_4_ with 5% methanol, pH 6.8 as a mobile phase was used. The eluent flow was 1 mL/min. The injection volume was 5–30 μL depending on signal intensity. We recorded 10 chromatograms from the absorption detector and 5 chromatograms from the fluorescence detector of each sample, which formed the basis for 3D chromatograms (Table 2). The excitation and emission wavelength of the fluorescence detector were selected according to our previous findings [13,14].

### 2.5. Data Processing and Statistical Analysis

Software LC solution (Shimadzu Co., Kyoto, Japan) was used for analysis and processing of chromatograms. Statistical analyses of data and graphing were performed using StatsDirect 3 (intergroup comparison, correlation, and receiver operating characteristic (ROC) analysis) and R software (logistic regression, partial least square discriminant analysis, orthogonal projection to latent structure discriminant analysis). OriginPro2016 software was used to create 3D chromatograms.

The areas under each chromatographic peak were normalized to urine creatinine area so that fluctuation in urine concentration was minimized. The creatinine identification was performed with absorption detection at 220 nm. For inter-group comparison, the *t*-test or the non-parametric alternative Mann–Whitney U test was used. To assess the predictive ability of the proposed diagnostic test, a ROC analysis was performed. The logistic regression was used to reveal the best classification model. Parameters for logistic regression models were selected according to the Akaike information criterion (AIC) and Log-likelihood function. Partial least square discriminant analysis (PLS-DA) and an orthogonal projection to latent structure discriminant analysis (OPLS-DA) models based on identified urine peaks were used to distinguish patients with BC from NMHU patients. The resulting model was verified using cross-validation, a permutation test, and external validation. Variable importance in projection (VIP) scores estimate the importance of each variable (peak) in the projection used in a PLS model.

## 3. Results

Ten chromatograms for the absorption detector and five chromatograms for the fluorescence detector were recorded from the urine of patients with BC and patients with NMHU (Figure 1). The total number of identified peaks was 334 at all detector settings in each urine sample. The peak of the sample was identified within the interval RT ± 6 s. Areas under all peaks were calculated and then normalized to the creatinine peak area. Normalized peak values were entered into further analyses.

Inter-group comparison of urine samples from patients with NMHU (*n* = 20) and patients with BC (*n* = 19) was performed and 18 statistically significant peaks (*p* < 0.05) were identified (6 peaks detected by the fluorescence detector, 12 peaks detected by absorption detector, Table 3 and Table 4).

### 3.1. ROC (Receiver Operating Characteristic) Analysis

The 18 peaks identified as significantly different for BC and NMHU patients (training set—19 from patients with BC and 20 from patients with NMHU) were individually entered into the ROC analysis to discriminate between BC and NMHU patients. The area under the ROC curve (AUC) represented the evaluation index. The highest value of AUC and best classification power was achieved by the fluorescent peak (370/520 nm; ex/em) identified with RT = 2.32 min (F (2.32 min)) with 90% sensitivity and 74% specificity (Table 5, Figure 2).

### 3.2. Logistic Regression Analysis

To identify an optimal and simplified biomarker set for distinguishing BC and NMHU patients, we performed the logistic regression analysis of the training set (*n* = 39). The 18 peaks were used as variables. The aim was to create a model with the best classification power by using the smallest possible number of peaks. Based on the AIC and Log-likelihood values, the best model for patient’s classification required 7 peaks (2 fluorescent and 5 absorption peaks): F(2.28 min); F(2.32 min); A(1.77 min); A(4.62 min); A(1.77 min); A(1.77 min); A(2.19 min) (Table 6). This classification model provided 100% sensitivity and 100% specificity.

### 3.3. Discriminant Analysis

The partial least square discriminant analysis (PLS-DA) was used to distinguish patients with BC and NMHU based on urinary metabolites (Figure 3). The model was created from a training set of samples (*n* = 39). All identified urine peaks by RP-HPLC (*n* = 334) were used for the model and it was built from two components. This model retains 33% of the variability from the original data. R2Y as a test prediction ability had a value of 86.2%. The goodness of prediction (prediction power) of the cross-validation test was 65.8% (test set of samples was used, *n* = 8). The permutation test was 0.05 (*p*-value) for pR2Y and pQ2.

We also performed an OPLS-DA analysis (Figure 4) to discriminate between BC and NMHU patients (training set of samples, *n* = 39). A model OPLS-DA was obtained with one predictive and two orthogonal components. The model retained 38.3% of the variability of the original data. The prediction rate of the test (separation ability of the test) was 91.6%, with cross-validation of 60.8% (test set of samples, *n* = 8). The permutation test was 0.05 for both pR2Y and pQ2. Potential biomarkers were selected based on the OPLS-DA model by variable importance in the projection score (VIP > 1, Figure 5).

Sequentially, external validation was performed on the test set consisting of eight patients (four BC patients and four NMHU patients). The model achieved 100% sensitivity and 80% specificity. In addition to external validation, we also classified all samples (training set + test set; *n* = 47) when the test sensitivity acquired 100% and specificity increased to 96%. The test identified one from 47 patients as false positive.

## 4. Discussion

The biggest challenge in bladder cancer (BC) diagnostics is to identify the disease before progression. Recently, there has been an increasing interest in non-invasive diagnostic tumor markers due to the invasive attribute of cystoscopy, which is still considered the gold standard diagnostic method [2]. However, markers published in the literature so far do not meet expectations for replacing cystoscopy due to their relatively low specificity and excessively high false-positive results, which can be mainly caused by frequently occurring hematuria [15]. Therefore, the decisive question for the urologists is to reliably and rapidly distinguish patients with bladder cancer from non-malignant hematuria (NMHU) patients. The urinary metabolomics-based diagnostic approach could have clinical relevance, because urine is in direct contact with bladder epithelial cells that may give rise to BC, and thus metabolites released from BC cells may be present in urine samples. Consistent with these findings, urine metabolomic analysis is a promising non-invasive approach for BC detection and marker discovery [8,9].

In this study, we focused on finding an appropriate set of biomarkers from the urinary metabolites of patients with BC and NMHU. For the separation and subsequent analysis of a large number of urine components, we chose a non-invasive, relatively fast, affordable, and analytically undemanding technique HPLC with absorption and fluorescence detection. The obtained dataset was further subjected to various uni- and multi-dimensional statistical analyses and mathematical modeling. The total number of identified untargeted chromatographic peaks was 334 in each urine sample, from which 18 peaks were significantly different for BC and NMHU patients (Table 3 and Table 4). These peaks represent urine metabolites with absorption and fluorescent abilities.

The 18 identified peaks were first subjected to a univariate ROC analysis. The best ROC analysis parameters (specificity of 74%, sensitivity of 90%) provided the fluorescent peak F(2.32 min) (Table 5).

Several other studies of univariate analysis of BC patients that included hematuria have used the BTA and NMP22 biomarkers [16,17,18]. The BTA stat test revealed a sensitivity of 72% to differentiate between BC and NMHU urine samples [18], which is lower than our classification with the biomarker F(2.32 min) and a specificity of 24–80% [16,18], which is comparable to our results of F(2.32 min). Another potential biomarker NMP22 BladderChek assay with specificity in the range 77–96% and sensitivity of 51–85% also exhibits false positivity for hematuria [17,19,20,21]. Compared to our results, we achieved higher sensitivity values, but worse specificity values over NMP22.

Considering none of the 18 identified peaks showed sufficient power to distinguish BC patients from NMHU, we further applied various multidimensional statistical analysis approaches on our multidimensional dataset of untargeted urine metabolites to identify and optimize the urine biomarker set.

Firstly, we performed a logistic regression analysis with the 18 peaks used as variables. Our model of the smallest possible number of peaks with the best classification power required seven peaks that correspond to the absorbing and fluorescent urine metabolites: A(1.77 min); A(4.62 min); A(1.77 min); A(1.77 min); A(2.19 min); F(2.28 min); and F(2.32 min) (Table 6). This model provided above average classification power to distinguish patients with BC from NMHU (100% sensitivity and 100% specificity).

Subsequently, the 334 chromatographic peaks of urine samples (training set) were subjected to partial least square discriminant analysis (PLS-DA; Figure 3) and orthogonal projection to latent structures discriminant analysis (OPLS-DA; Figure 4) for classification between BC and NMHU patients. By external validation (test set), we were able to distinguish BC patients from NMHU patients with sensitivity and specificity of 100% and 80%. By testing the classification of all patients (training set + test set *n* = 47) into diagnostic groups, the differential model achieved 100% sensitivity and 96% specificity. Jin et al. [22] used high-performance liquid chromatography-quadrupole time-of-flight mass spectrometry (HPLC-QTOFMS) to perform urine metabolomic profiles of BC patients and control group, which included healthy subjects and hematuria patients. Their OPLS-DA analysis afforded sensitivity of 91.3% and specificity of 92.5% and PLS-DA-based ROC curve analysis achieved 85% sensitivity and specificity. Compared to our discriminant analyses, we achieved better results of both sensitivity and specificity.

## 5. Conclusions

Our work examined the possibilities of non-targeted biomarkers of urine to distinguish patients with malignant and nonmalignant diseases of the bladder using 3D HPLC in combination with computer processing of multiple datasets (334 chromatographic peaks in each sample). By logistic regression, we identified an optimal and simplified set of seven chromatographic peaks (five absorptions plus two fluorescence) with strong classification power (100% sensitivity and 100% specificity) for distinguishing patients with bladder cancer and those with non-malignant hematuria. The differentiation model (OPLS-DA) diagnosed BC with a sensitivity and specificity of 100% and 96%. Monitoring chromatographic peaks with absorption and fluorescent detection thus showed potential for the non-invasive diagnostic test, that can initially and rapidly distinguish patients with malignant and non-malignant hematuria. In addition, the use of this method is fast and inexpensive, requires minimal sample preparation, and, according to our results, has the potential to achieve high accuracy. Prospectively, the proposed method could be an accessible ambulatory tool for diagnostics, therapeutic progress, and recurrence prevention of bladder cancer.

## Figures and Tables

**Figure 1 diagnostics-11-01793-f001:**
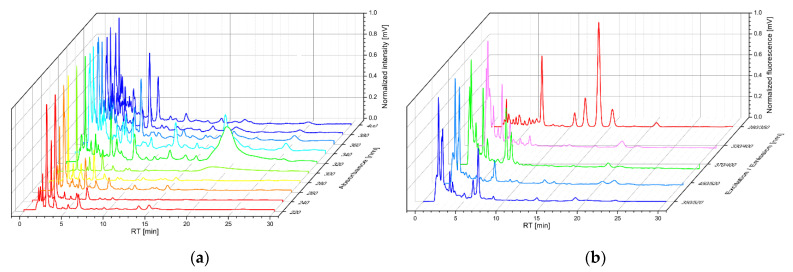
Representative normalized 3D chromatograms of urine for (**a**) absorption detector, (**b**) fluorescence detector; RT —retention time.

**Figure 2 diagnostics-11-01793-f002:**
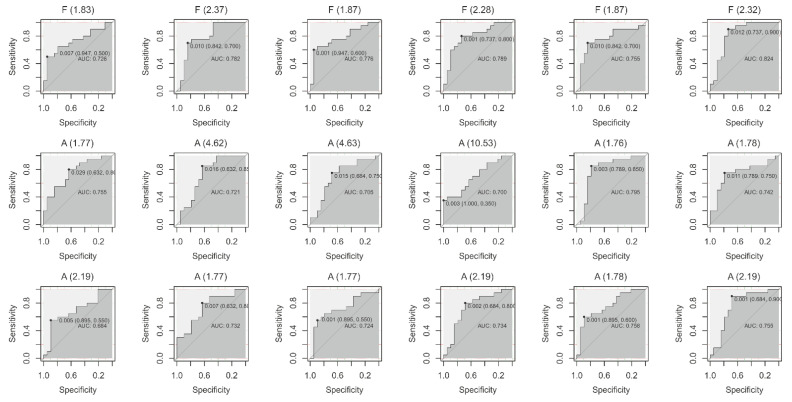
ROC curves for 18 significant peaks.

**Figure 3 diagnostics-11-01793-f003:**
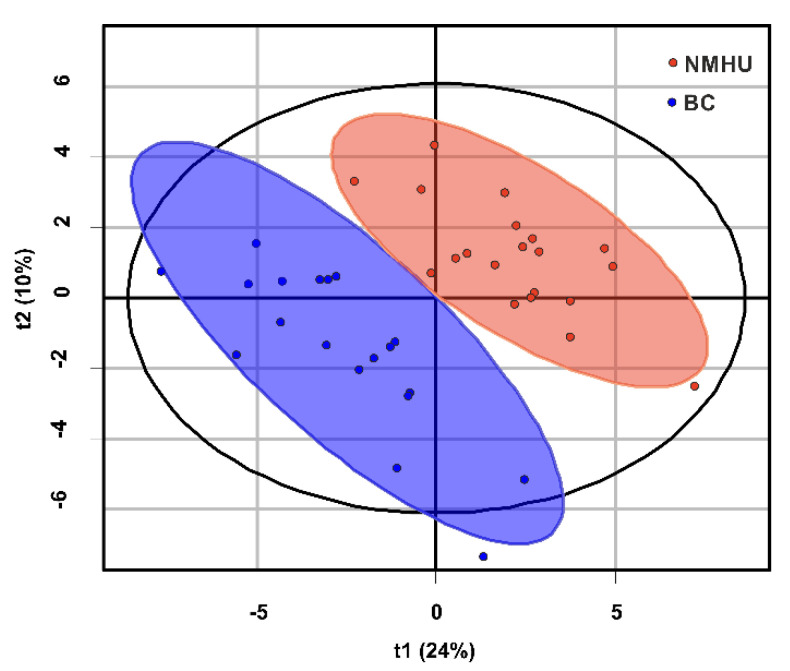
PLS-DA score plot (patients with bladder cancer (blue), patients with non-malignant hematuria (red); dots represent the metabolomics profile of the urine of an individual subject).

**Figure 4 diagnostics-11-01793-f004:**
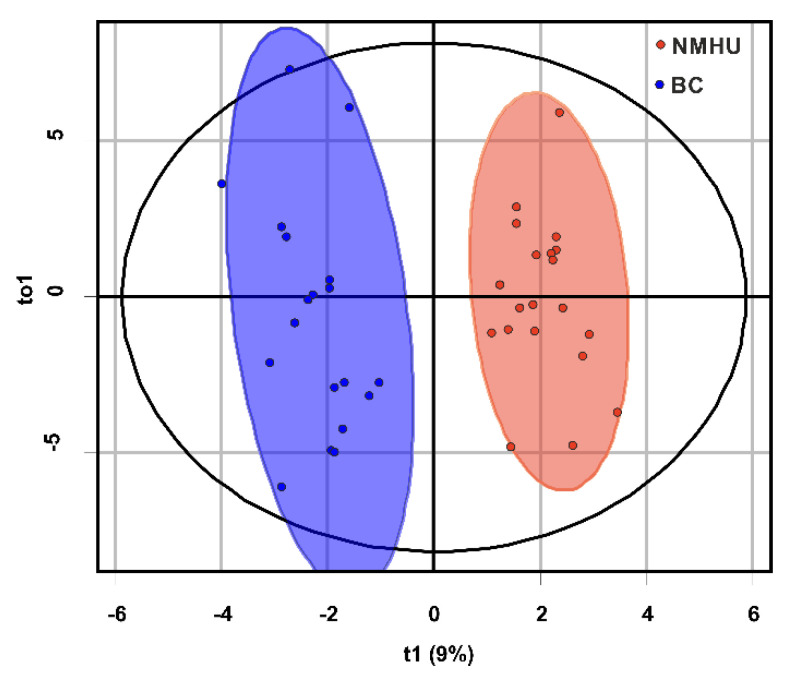
OPLS-DA score plot (patients with bladder cancer (blue), patients with non-malignant hematuria (red); dots represent the metabolomics profile of the urine of an individual subject).

**Figure 5 diagnostics-11-01793-f005:**
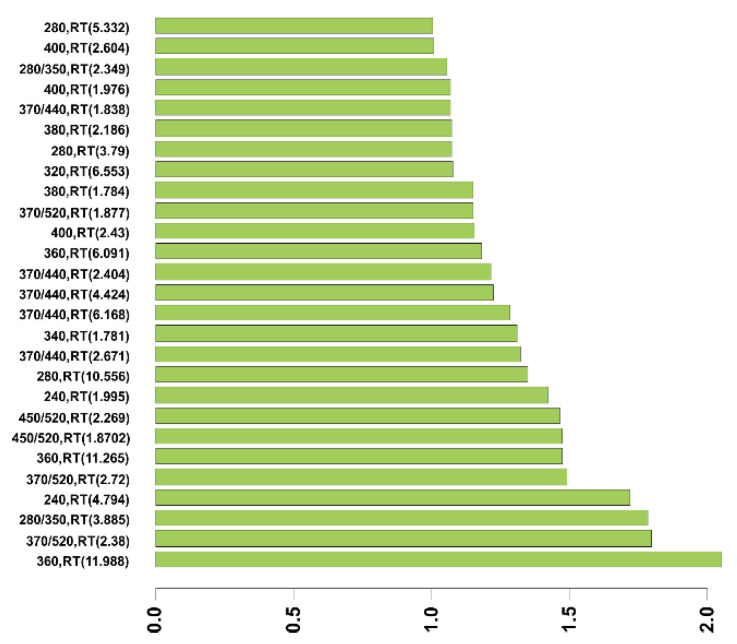
Characteristics of peaks for group differentiation according to the OPLS-DA model (variable importance in projection, VIP >1). The y-axis represents the peak designation (absorption, excitation/emission wavelength (nm), RT—retention time (min)).

**Table 1 diagnostics-11-01793-t001:** Clinicopathological characterization of patients.

Characteristics	Non-Malignant Hematuria (NMHU)	Bladder Cancer (BC)
Age (mean ± SD)	65.2 ± 8.2	65.4 ± 5.4
No. of subject	24	23
Training set	20	19
Test set	4	4
Cancer Grade		
G1	-	11
G2	-	3
G3	-	9
Cancer Stage		
Ta	-	10
T1	-	9
T2	-	4

SD—standard deviation, G1—well-differentiated tumor (low grade), G2—moderately differentiated tumor (intermediate grade), G3—poorly differentiated tumor (high grade), Ta—non-invasive tumor (only in the innermost layer), T1—solitary tumor without muscular invasion, T2—solitary tumor with muscular invasion.

**Table 2 diagnostics-11-01793-t002:** Wavelength settings of absorption and fluorescence detectors.

	Wavelength (nm)									
Absorption	220	240	260	280	300	320	340	360	380	400
Excitation/ Emission	280/350		330/400		370/440		450/520		370/520	

**Table 3 diagnostics-11-01793-t003:** Characteristics of fluorescent peaks identified as significantly different for BC and NMHU groups.

Excitation/Emission (Wavelength (nm))	*p*-Value	RT (min) (SD)	Area/Area of Creatinine × 10^−3^ Median (IQR)	
			Non-Malignant Hematuria (NMHU)	Bladder Cancer (BC)
370/440	0.0151	1.83	8.13	12.96
		(0.012)	(5.03–14.35)	(11.47–18.83)
370/440	0.0021	2.37	8.43	11.98
		(0.036)	(6.17–11.06)	(10.25–18.13)
450/520	0.0026	1.87	0.93	1.98
		(0.013)	(0.75–1.92)	(1.36–3.50)
450/520	0.0015	2.28	0.78	1.46
		(0.008)	(0.55–1.21)	(1.07–2.23)
370/520	0.0057	1.87	7.84	14.63
		(0.014)	(6.04–13.32)	(10.18–17.24)
370/520	0.0002	2.32	9.21	15.80
		(0.051)	(7.80–10.93)	(11.29–20.23)

RT—retention time, SD—standard deviation, IQR—interquartile range.

**Table 4 diagnostics-11-01793-t004:** Characteristics of absorption peaks identified as significantly different for BC and NMHU groups.

Wavelength (nm)	*p*-Value	RT (min) (SD)	Area/Area of Creatinine × 10^−3^ Median (IQR)	
			Non-Malignant Hematuria (NMHU)	Bladder Cancer (BC)
240	0.0057	1.77	26.09	31.75
		(0.024)	(22.82–29.17)	(26.50–39.79)
240	0.0177	4.62	10.95	17.01
		(0.033)	(8.10–14.24)	(9.85–46.14)
260	0.0283	4.63	12.70	17.91
		(0.037)	(8.47–15.85)	(13.46–25.52)
280	0.328	10.53	3.46	4.82
		(0.045)	(2.36–4.89)	(3.09–9.83)
320	0.0012	1.76	2.15	3.48
		(0.025)	(1.78–2.64)	(2.75–4.31)
340	0.0090	1.78	7.90	13.18
		(0.022)	(6.62–11.50)	(10.73–14.78)
340	0.0501	2.19	4.88	9.36
		(0.024)	(4.16–10.29)	(5.72–10.90)
360	0.0128	1.77	4.66	7.38
		(0.020)	(3.07–6.37)	(4.86–10.10)
380	0.0164	1.77	1.41	2.75
		(0.019)	(1.01–2.91)	(1.87–4.58)
380	0.0087	2.19	1.25	1.74
		(0.007)	(1.08–1.54)	(1.30–2.36)
400	0.0052	1.78	0.89	1.60
		(0.019)	(0.89–0.89)	(0.92–2.88)
400	0.0057	2.19	0.72	1.08
		(0.007)	(0.64–0.93)	(0.79–1.28)

RT—retention time, SD—standard deviation, IQR—interquartile range.

**Table 5 diagnostics-11-01793-t005:** Summary of ROC curve analysis of 18 significant peaks.

Peak (RT (min))	Wavelength (ex/em (nm))	Cut-Off Value	Specificity (%) (CI)	Sensitivity (%) (CI)	NPV (%)	PPV (%)	AUC
F (1.83)	370/440	0.007	94.7	50	64	91	0.726
			(84.2–100)	(30–75)			
F (2.37)	370/440	0.010	84.2	70	73	82	0.782
			(68.4–100)	(50–90)			
F (1.87)	450/520	0.001	94.7	60	69	92	0.776
			(84.2–100)	(40–80)			
F (2.28)	450/520	0.001	73.7	80	78	76	0.789
			(52.6–94.7)	(60–95)			
F (1.87)	370/520	0.010	84.2	70	73	82	0.755
			(68.4–100)	(50–90)			
F (2.32)	370/520	0.012	73.7	90	88	78	0.824
			(52.6–89.5)	(75–100)			
A (1.77)	240	0.029	63.2	80	75	70	0.755
			(42.1–84.2)	(60–95)			
A (4.62)	240	0.016	63.2	85	80	71	0.721
			(42.1–84.2)	(69.9–100)			
A (4.63)	260	0.015	68.4	75	72	71	0.705
			(47.4–89.5)	(55–90)			
A (10.53)	280	0.003	100	35	59	100	0.700
			(100–100)	(15–55)			
A (1.76)	320	0.003	78.9	85	83	81	0.795
			(57.9–94.7)	(70–100)			
A (1.78)	340	0.011	78.9	75	75	79	0.742
			(57.9–94.7)	(55–95)			
A (2.19)	340	0.005	89.5	55	65	85	0.684
			(73.7–100)	(35–80)			
A (1.77)	360	0.007	63.2	80	75	70	0.732
			(42.1–84.2)	(60–95)			
A (1.77)	380	0.001	89.5	55	65	85	0.724
			(73.7–100)	(30–75)			
A (2.19)	380	0.002	68.4	80	76	73	0.734
			(47.4–89.5)	(60–95)			
A (1.78)	400	0.001	89.5	60	68	86	0.758
			(73.7–100)	(35–80)			
A (2.19)	400	0.001	68.4	90	87	75	0.755
			(47.4–89.5)	(75–100)			

NPV—negative predictive value, PPV—positive predictive value, AUC—area under ROC curve, RT—retention time, A—absorption detector, F—fluorescence detector, ex/em—excitation/emission, CI—95% confidence interval.

**Table 6 diagnostics-11-01793-t006:** Results of models obtained from logistic regression.

Peak (RT [min])	F(1.83)	F(2.37)	F(1.87)	F(2.28)	F(1.87)	F(2.32)	A(1.77)	A(4.62)	A(4.63)	A(10.53)	A(1.76)	A(1.78)	A(2.19)	A(1.77)	A(1.77)	A(2.19)	A(1.78)	A(2.19)	Sensitivity (%)	Specificity (%)	Log-likelihood	AIC
0																					27.5	
1						■													74	85	35.07	41.83
2						■		■											89	90	39.80	33.70
3						■	■	■											84	90	41.65	33.70
4						■		■						■	■				84	90	43.70	31.14
5	■					■		■						■	■				89	100	45.65	28.53
6				■	■	■		■			■							■	95	95	47.61	23.00
7				■		■	■	■						■	■	■			**100**	**100**	**50.78**	**16.00**

■—peak that entered the model, bold—best model for diagnostic test, AIC—Akaike information criterion, RT—retention time, F—fluorescence detector, A—absorption detector.

## Data Availability

Data available upon reasonable request to the corresponding author.

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
