# Peer review of "Multiple Chromatographic Analysis of Urine in the Detection of Bladder Cancer"

_diagnostics, 2021, doi:10.3390/diagnostics11101793_

Round 1
Reviewer 1 Report
In the present study, the authors have attempted to perform an untargeted metabolomics approach using 3D reverse-phase high-performance liquid chromatography (RP-HPLC) with absorption and fluorescence detection, which was carried out as an alternative to omics methods for search bi-omarkers in patient’s urine. They have also used computer processing of acquired extensive data (Logistic regression, Receiver Operating Characteristic (ROC) analysis, Partial Least Square Discriminant Analysis, Orthogonal Projection to Latent Structure Discriminant Analysis), and thus they have created a classification models for the differentiation of patients with bladder cancer and (BC) and Non-malignant Hematouria (NMHU).
From one perspective the authors have attempted to assess the use of the HPLC method in the detection (early?) of BC. The us eof urine in the diagnosis of BC is broadly used. For example, the cytologic examination of urine samples for malignant cells is a very common method used in cases of suspicion for BC. In that sense, the authors should address what is the additive value of their methodological approach. Especially, when HPLC is not an equipment that is readily available in the majority of laboratories.
From a methodological perspective, the authors have frozen the urine samples and thawed later. This would cause the rupture of the red blood cells (RBC) and the subsequent release of hemoglobin and cell membrane fragments into the urine solution. How did they account for this in their analysis? Based, also, on my previous comment, in a sample with ruptured cells it would be more reasonable to use a PCR or RT-PCR method in order to detect the malignant cells with a molecular method.
As they also state, the presence of hematouria is a sign of late cancer detection. how is their approach to assist towards the early detection of BC?
To my understanding the authors chose to assess the performance of their method using the creatinine peaks. However, why go into the trouble to detect creatinine, which is easily detected with other more easy methods, using HPLC? In addition, the authors present the obtained spectra from the examined sample, yet they do not use this information for the detection of BC. For example, why not compute the area under the identified spectra and compare the BC samples to NMHU? please elaborate on that. It is not clear if the data presented in Tabled 5 and 6 concern the sum of peaks or the aforementioned creatinine peaks. In addition, it would more helpful for the reader to see also some actual ROC curves.
In addition, please present the spectra with some identification. In particular, what is that we see in figure 1? Enhance the legends, as well as mark on the figure some info in order to understand its scope. Which spectra belong to the patients, which to the NMHU? where is the creatinine peak?
Further on, the identified peaks (18) correspond to what? how did they come up with the statistical significance between BC and NMHU? Please make this very clear.
Not all readers are familiar with the HPLC method, thus it would be useful to present some brief background in order to make the results more clear.
Finally, the authors shouldf summarize their findings, highlight them and add the significance of their approach in the detection of BC. Is it more precise, more rapid, has prognostic value?
The figure
Reviewer 2 Report
Dzubinska et al. performed 3D HPLC analysis of the urine from patients with bladder cancer (BC) and non-malignant hematuria (NMHU). They found 334 chromatographic peaks, of which 18 peaks were significantly different between BC and NMHU. They found 7 chromatographic peaks with strong classification power by logistic regression. They further performed multivariate analysis of urinary metabolomic profile by PLA-DA and OPLA-DA to distinguish between BC and NMHU, and they successfully discriminated BC and NMHU.
This is an interesting and potentially useful study for non-invasive diagnosis of BC.
I raise some points that should be revised.
- Table 1, caption: T1, without muscular invasion and T2, with muscular invasion. Not vascular invasion.
- (p.5, l.7) with RT-2.38 min (F(2.38)): According to Table 5, these must be 2.32.
- (p.6, ll.6-8 & p.9, ll.27-30) 7 peaks that correspond to the absorption and fluorescence urine metabolite: A(1.78 m), A(2.19 m), A(1.77 m), A(4.59 m), A (1.78 m), F (2.27 m), F(2.38 m). However, these must be F(1.87), F(1.87), F(2.32), A(1.77), A(1.77), A(1.77), A(2.19) according to the Table 6, row 7.
- (p.9, l.14) peak F(2.38 min): According to Table 5, these must be 2.32.
Author Response
Dzubinska et al. performed 3D HPLC analysis of the urine from patients with bladder cancer (BC) and non-malignant hematuria (NMHU). They found 334 chromatographic peaks, of which 18 peaks were significantly different between BC and NMHU. They found 7 chromatographic peaks with strong classification power by logistic regression. They further performed multivariate analysis of urinary metabolomic profile by PLA-DA and OPLA-DA to distinguish between BC and NMHU, and they successfully discriminated BC and NMHU.
This is an interesting and potentially useful study for non-invasive diagnosis of BC.
I raise some points that should be revised.
- Table 1, caption: T1, without muscular invasion and T2, with muscular invasion. Not vascular invasion.
Corrected.
- (p.5, l.7) with RT-2.38 min (F(2.38)): According to Table 5, these must be 2.32.
Corrected. First, the peaks were named according to the reference time, then so that it would not be more confusing, we named the peaks according to the average time. But we didn't fix it in the whole text. Thank you for your comment.
- (p.6, ll.6-8 & p.9, ll.27-30) 7 peaks that correspond to the absorption and fluorescence urine metabolite: A(1.78 m), A(2.19 m), A(1.77 m), A(4.59 m), A (1.78 m), F (2.27 m), F(2.38 m). However, these must be F(1.87), F(1.87), F(2.32), A(1.77), A(1.77), A(1.77), A(2.19) according to the Table 6, row 7.
Corrected. (Same reason)
- (p.9, l.14) peak F(2.38 min): According to Table 5, these must be 2.32.
Corrected. (Same reason)